# Comorbidities Across Functional Neurological Disorder Subtypes: A Comprehensive Narrative Synthesis

**DOI:** 10.3390/life15081322

**Published:** 2025-08-20

**Authors:** Ioannis Mavroudis, Katerina Franekova, Foivos Petridis, Alin Ciobîca, Dăscălescu Gabriel, Emil Anton, Ciprian Ilea, Sotirios Papagiannopoulos, Dimitrios Kazis

**Affiliations:** 1Department of Neuroscience, Leeds Teaching Hospital, NHS Trust, Leeds LS17 7HY, UK; ioannis.mavroudis@gmail.com; 2Institute of Health Sciences, University of Leeds, Leeds LS2 9NL, UK; katarina.franekova@qcentrum.sk; 3Third Department of Neurology, Aristotle University of Thessaloniki, 54124 Thessaloniki, Greece; f_petridis83@yahoo.gr (F.P.); spapagia@auth.gr (S.P.); dimitrios.kazis@gmail.com (D.K.); 4Academy of Romanian Scientists, 050085 Bucharest, Romania; 5Department of Biology, Faculty of Biology, “Alexandru Ioan Cuza” University of Iasi, 700506 Iasi, Romania; alin.ciobica@uaic.ro (A.C.); gabidascalescu2001@gmail.com (D.G.); 6CENEMED Platform for Interdisciplinary Research, University of Medicine and Pharmacy “Grigore T. Pop”, 700115 Iasi, Romania; 7“Olga Necrasov” Center, Department of Biomedical Research, Romanian Academy, 010071 Iasi, Romania; 8“Ioan Haulica” Institute, Apollonia University, 700511 Iasi, Romania; 9Faculty of Medicine, University of Medicine and Pharmacy “Grigore T. Pop”, 700115 Iasi, Romania; cilea1979@yahoo.com

**Keywords:** functional neurological disorder, FND subtypes, psychiatric comorbidities, somatic comorbidities, FMD, FCD

## Abstract

Background: Functional Neurological Disorder (FND) encompasses a spectrum of symptoms—including motor, cognitive, and seizure-like manifestations—that are not fully explained by structural neurological disease. Accumulating evidence suggests that comorbid psychiatric and somatic conditions significantly influence the clinical course, diagnostic complexity, and treatment response in FND. Objective: This study systematically explores psychiatric and medical comorbidities across major FND subtypes—Functional Cognitive Disorder (FCD), Functional Movement Disorder (FMD), and Psychogenic Non-Epileptic Seizures (PNES)—with an emphasis on subtype-specific patterns and shared vulnerabilities. Methods: We conducted a narrative review of the published literature, guided by systematic principles for transparency, covering both foundational and contemporary sources to examine comorbid conditions in patients with FCD, FMD, PNES, PPPD and general (mixed) FND populations. Relevant studies were identified through structured research and included based on methodological rigor and detailed reporting of comorbidities (PRISMA). Extracted data were organized by subtype and comorbidity type (psychiatric or medical/somatic). Results: Across all FND subtypes, high rates of psychiatric comorbidities were observed, particularly depression, anxiety, PTSD, and dissociative symptoms. FCD was predominantly associated with internalizing symptoms, affective misattribution, and heightened cognitive self-monitoring. FMD demonstrated strong links with trauma, emotional dysregulation, and personality vulnerabilities. PNES was characterized by the highest burden of psychiatric illness, with complex trauma histories and dissociation frequently reported. Somatic comorbidities—such as fibromyalgia, chronic pain, irritable bowel syndrome, and fatigue—were also prevalent across all subtypes, reflecting overlapping mechanisms involving interoception, central sensitization, and functional symptom migration. Conclusions: Comorbid psychiatric and medical conditions are integral to understanding the presentation and management of FND. Subtype-specific patterns underscore the need for individualized diagnostic and therapeutic approaches, while the shared biopsychosocial mechanisms suggest benefits of integrated care models across the FND spectrum.

## 1. Introduction

Functional Neurological Disorder (FND) is a condition in which patients experience neurological symptoms (such as limb weakness, tremors, seizures, or cognitive difficulties) without a detectable structural abnormality in the nervous system. These symptoms are genuine and involuntary, and they are thought to arise from disruptions in brain functions, particularly in how the brain processes sensory and motor signals [1].

FND encompasses several clinical subtypes, including Functional Movement Disorder (FMD), characterized by tremors, weakness, or abnormal gait; Psychogenic Non-Epileptic Seizures (PNES), marked by seizure-like episodes without epileptic activity; Functional Cognitive Disorder (FCD), where patients report memory and attention difficulties not consistent with dementia, and Persistent Postural-Perceptual Dizziness (PPPD), involving chronic dizziness without a vestibular case. Although the exact etiology of FND remains incompletely understood, several interacting factors have been proposed. These include psychological stressors, trauma (especially early-life adversity), maladaptive cognitive styles, and alterations in attention or perception. Emerging research also points to possible genetic predispositions, abnormalities in neural network functioning, as well as, in rare cases, potential post-infectious or autoimmune triggers. The interplay between biological vulnerability, psychological factors, and social context supports the current biopsychological model of FND [2]. Early descriptions of FND-like presentation date back centuries, but it was Jean-Martin Charcot who, in the late 19th century, first underscored its clinical significance alongside other neurological and psychiatric disorders [3]. Throughout much of the 20th century, however, FND was relegated to the margins of medical practice as diagnostic technologies lagged, and the disciplines of neurology and psychiatry became increasingly siloed.

Over the past three decades, a renaissance in FND research has been driven by advances in diagnostic criteria, high-resolution neuroimaging, and randomized controlled trials of biopsychosocial interventions. This paradigm shift has moved the field beyond purely psychodynamic explanations, toward an integrative model in which biological predispositions, psychological processes, and sociocultural context interact to produce and maintain FND symptoms [4,5,6]. Today, FND is recognized as the second most common diagnosis in outpatient neurology clinics, second only to headache disorders, underscoring both its clinical burden and the urgent need for enhanced therapeutic strategies [5,7].

Epidemiological data reveal substantial variability in FND incidence and prevalence across settings. In neurology referral cohorts, FND accounts for approximately 5–10% of new consultations, while population-based estimates in the United Kingdom suggest an annual incidence of about 12 per 100,000 individuals, equating to roughly 8000 new cases each year and a community prevalence of 50,000–100,000 people [1]. Given diagnostic complexity and limited specialist access, these figures likely underestimate the true scope of FND worldwide.

Although FND is most commonly present in young to middle adulthood, burgeoning evidence highlights its relevance in pediatric populations as well. In a 36-month cohort study conducted by Yong et al. (2023) at a regional children’s hospital, the incidence of FND was 18.3 per 100,000 children, with a median age of onset of 13 years and a pronounced female predominance (70%) [8]. This gender disparity aligns with a one-stage meta-analysis by Lidstone et al. (2023), which found that approximately 70% of FND cases, across motor, sensory, and seizure-like phenotypes, occur in female patients [9].

Geographic and socioeconomic factors further shape FND epidemiology. Higher rates are reported in industrialized nations compared to low- and middle-income countries, likely reflecting differences in healthcare infrastructure, diagnostic practices, and cultural attitudes toward neuropsychiatric illness [10,11]. Moreover, social determinants such as lower education attainment, financial insecurity, and limited social support have been associated with increased vulnerability to FND, emphasizing the importance of contextual factors onset and maintenance of the disorder.

From a clinical standpoint, it is important to recognize that symptom expression, healthcare access, and treatment adherence in FND may vary across different ethnic and cultural backgrounds. Populations such as African American, Pacific Islander, Asian, Caucasian, and European heritage groups may differ in illness beliefs, stigma, and engagement with multidisciplinary care. However, our review identified very limited research directly comparing comorbidity profiles or treatment responses across these groups. This lack of data represents a limitation in the current evidence base and underscores the need for culturally sensitive, population-specific research.

Despite its prevalence and impact, the neurobiology of FND remains incompletely elucidated, and distinguishing genuine functional symptoms from malingering or feigned presentations continues to challenge clinicians and contribute to stigmatization [4,5,6].

Functional neuroimaging studies have implicated aberrant activity and connectivity within limbic, salience, agency, and sensorimotor networks, supporting the conceptualization of FND as a disorder of neural network dysfunction rather than overt structural pathology [5].

While substantial progress has been made in refining diagnostic frameworks and elucidating pathophysiological mechanisms, comparatively few studies have systematically characterized the spectrum of psychiatric and medical comorbidities that accompany FND. Such comorbidities frequently influence symptom expression, treatment response, and long-term prognosis, yet their prevalence and impact are inconsistently reported across the literature.

The objective of this narrative review is to synthesize current evidence on the comorbid conditions associated with FND and its principal subtypes. We will examine the prevalence and nature of psychiatric and somatic comorbidities across diverse FND phenotypes, identify clinically relevant patterns, discuss implications for diagnostic formulation and multidisciplinary management, and highlight key gaps in existing research. By mapping these comorbidity profiles, we aim to foster a more integrated, patient-centered approach to FND that can inform both clinical practice and future investigations.

## 2. Materials and Methods

### 2.1. Study Design

This paper is structured as a comprehensive narrative review aiming to synthesize evidence regarding comorbidities in FND. Although a PRISMA flow diagram was included to transparently report the article selection process, this is not a systematic review in the strict methodological sense. Instead, we followed structured search and inclusion criteria to ensure methodological rigor while maintaining the broader analytical flexibility typical of narrative syntheses.

We conducted a narrative review of peer-reviewed literature to explore psychiatric and somatic/medical comorbidities in Functional Neurological Disorder (FND) and its principal subtypes, psychogenic non-epileptic seizures (PNES), persistent-postural-perceptual dizziness (PPPD), functional movement disorders (FMD), and functional cognitive disorder (FCD). Our goal was to synthesize evidence on comorbidity prevalence and character to inform holistic clinical management and guide future investigation.

### 2.2. Search Strategy and Data Source

Three electronic databases (PubMed, Scopus and Web of Science) were searched through January 2025 using a comprehensive Boolean query that combined FND-related terms (“functional neurological disorder”, “conversion disorder”, “PNES”, “PPPD”, “FMD”, and “FCD”) with comorbidity-related keywords (“comorbidity”, “depression”, “anxiety”, “PTSD”, “personality disorder”, “fibromyalgia”, “migraine”, “IBS”, etc.). We restricted results to English-language, peer-reviewed articles. The included literature spans from early foundational works to the most recent peer-reviewed publications available as of January 2025.

### 2.3. Eligibility Criteria

We included studies that reported original clinical, epidemiological, or structured-review data on adults or mixed-age cohorts diagnosed with FND or one of its subtypes, explicitly examining psychiatric (e.g., mood and anxiety disorders, PTSD, and personality disorders) or somatic/medical (e.g., fibromyalgia, migraine, chronic pain, and irritable bowel disease) comorbidities. We excluded pediatric-only investigations without a comorbidity focus, as well as editorials, opinion pieces, or any publications lacking systematic data.

### 2.4. Data Extraction and Classification

Two investigators independently extracted publication details, targeted FND phenotype(s), sample characteristics, and all reported comorbid conditions. Comorbidities were classified in three categories:-Psychiatric: depression, anxiety, PTSD, mood and personality disorders;-Somatic/Medical: fibromyalgia, migraine, IBS, chronic pain syndromes;-Unspecified: studies indicating relevance of FND without clear comorbidity details.

Studies were grouped by FND subtype, PNES, PPPD, FMD, FCD, or General FND, based on explicit mentions in titles, abstracts, or keywords.

### 2.5. Synthesis Approach

Through qualitative thematic synthesis, findings were organized into two integrative matrices: one mapping studies by comorbidity category and another aligning each FND subtype with its associated comorbidities. This narrative framework facilitated identification of recurring prevalence patterns, subtype-specific associations, and implications for clinical diagnosis and multidisciplinary management. Methodological and outcome heterogeneity precluded meta-analytic pooling.

## 3. Results

The initial search provided 354 studies (PubMed—143; Scopus—121; Web of Science —90). After excluding duplicates, 184 studies underwent title and abstract screening. Of these, 67 were excluded for not meeting the inclusion criteria. The remaining 117 studies progressed to full-text validation, all of which met eligibility criteria and were included in the final narrative synthesis (Figure 1). This review was not registered in PROSPERO, as data extraction and analysis were completed prior to the decision to register the protocol.

These 117 studies provided data on psychiatric and medical comorbidities associated with Functional Neurological Disorder subtypes, including psychogenic non-epileptic seizure (PNES), functional movement disorder (FMD), functional cognitive disorder (FCD), persistent postural-perceptual dizziness (PPPD), and general FND presentations. The following narrative synthesis details comorbidity patterns across subtypes, reporting patient-level prevalence percentages from included studies.

To highlight the heterogeneity of psychiatric and somatic comorbidities across FND subtypes, Table 1 presents a comparative synthesis of clinical patterns identified in the literature.

### 3.1. Psychogenic Non-Epileptic Seizures (PNES)

PNES represents one of the most clinically complex forms of FND, characterized by seizure-like episodes without corresponding EEG abnormalities. These episodes are highly comorbid with psychiatric disorders, trauma histories, and somatic symptom burden.

#### 3.1.1. Psychiatric Comorbidities

Psychiatric disorders emerged as the most prevalent and well studied comorbidities in PNES, reported in 44 of the 54 included studies. These included depression, anxiety, PTSD, dissociative disorders, and personality pathology, often predating the onset of seizures and influencing their course.

Several reviews described how mood and anxiety disorders frequently co-occur with PNES, leading to worse functional outcomes and complicating treatment strategies [13,14,15,16,17,18,19,20,21,22]. Depression and generalized anxiety disorder were the most cited, with many studies indicating these conditions are under-recognized in neurological settings [23,24,25].

Studies such as LaFrance et al. and Popkirov et al. [26,27] emphasized the role of early trauma and PTSD, suggesting these may act as both predisposing and perpetuating factors for PNES. Trauma-related dissociation and stress-related disorders are frequently featured in diagnostic interviews, often uncovered only during psychiatric evaluations.

Personality disorders were explored in depth in multiple studies. Traits associated with borderline, avoidant, and obsessive-compulsive features were frequently reported [26,27,28,29,30,31]. These personality characteristics were linked with emotional dysregulation, interpersonal difficulties, and poor coping mechanisms—all of which contribute to PNES persistence.

In neurobiological studies, such as those by Perez et al. and Gilmour et al. [13,32], shared neural circuits involving emotion regulation, threat detection, and self-agency were implicated in both psychiatric disorders and PNES. These findings support a model in which psychiatric symptoms are not merely coexisting, but neurobiologically intertwined with the pathophysiology of functional seizures.

Furthermore, psychiatric comorbidities were found to negatively influence treatment outcomes. Studies by Carson, Stone, and colleagues [33] showed that untreated psychiatric illness significantly lowers the chances of remission. They called for integrated, multidisciplinary care pathways including psychiatrists and psychologists.

Emerging perspectives also advocate for early screening and trauma-focused interventions [34,35,36,37]. These can help mitigate the escalation of symptoms and reduce diagnostic delays, which, as shown in the literature, often exceed several years in cases where psychiatric assessments were deferred [38,39].

#### 3.1.2. Somatic and Pain-Related Comorbidities

Somatic and pain-related comorbidities were discussed in 11 of the reviewed studies, focusing on chronic pain, fibromyalgia, fatigue syndromes, gastrointestinal disturbances, and functional sensory symptoms.

These studies highlighted that many PNES patients meet diagnostic criteria for central sensitivity syndromes, suggesting overlapping mechanisms of neural amplification and altered sensory processing [27,32,33,34]. For example, patients with fibromyalgia or chronic widespread pain often reported a history of PNES, and vice versa.

Carle-Toulemonde et al. [40] provided a compelling review linking functional somatic syndromes with functional neurological disorders, describing how interoceptive distortion and heightened autonomic responsivity may serve as common threads.

Additionally, several articles described how the burden of pain often obscures the diagnosis of PNES, as patients are referred repeatedly for physical rather than neuropsychiatric evaluations [20,21,41,42]. This diagnostic overshadowing contributes to significant health service utilization and unnecessary medical testing.

Some studies also discussed functional motor symptoms, such as gait disturbances and weakness, which often coexist with pain complaints and non-epileptic seizures [41,43]. These complex overlaps may reflect systemic vulnerability in stress-processing systems involving the insula, anterior cingulate cortex, and somatosensory integration areas [13,42].

Collectively, these studies underscore the need for coordinated evaluation of both neurological and somatic symptoms in patients presenting with seizure-like episodes.

#### 3.1.3. Cognitive Comorbidities

Cognitive symptoms were addressed in eight studies, with a focus on attention, executive function, subjective memory complaints, and functional cognitive disorder (FCD).

While objective cognitive impairment was often minimal or inconsistent, subjective complaints were frequent and significantly distressing [15,24,37,40]. These included difficulties with concentration, decision-making, language fluency, and memory retention.

In some cases, cognitive dysfunction was interpreted as part of a broader functional symptom spectrum, possibly reflecting dysfunction in attentional control networks. Functional imaging studies described altered connectivity in frontal and parietal networks responsible for top-down modulation of behavior [13,44,45].

Notably, patients often reported fear of neurodegeneration, prompting referrals for dementia screening, especially in older adults [31,39]. However, neuropsychological assessments generally fail to confirm neurodegenerative patterns, reinforcing the functional nature of complaints.

Two studies explicitly examined the link between cognitive symptoms and trauma history, suggesting that dissociative amnesia and trauma-related intrusions may underline some of the episodic memory difficulties in PNES [44,46].

Importantly, studies like those by Stone et al. and Gilmour et al. recommend that functional cognitive symptoms be identified early, as they significantly impact rehabilitation potential and patient engagement [31,32].

### 3.2. Functional Movement Disorder (FMD)

Motor Functional Neurological Disorder (FMD) is a prevalent and complex subtype of functional neurological disorders, characterized by abnormal movements such as tremors, dystonia, weakness, and gait disturbances without a structural neurological basis. Across 20 studies, the literature highlighted the clinical phenomenology, diagnostic strategies, neurobiological correlates, and treatment approaches of FMD, revealing consistent themes and evolving evidence.

#### 3.2.1. Clinical Features and Phenomenology

Several studies focused on describing the varied motor presentations of FMD, with tremor, dystonia, and functional weakness being the most common [47,48,49]. Perez et al. provided a detailed framework for neuropsychiatric assessment of motor symptoms, emphasizing the distinction between functional and organic movement disorders based on clinical signs, variability, and incongruence with neurological syndromes [47].

Sojka et al. explored patient experiences and illness beliefs in FMD, noting high levels of symptom chronicity, distress, and disability [13]. A recurring theme was the discrepancy between subjective symptom severity and objective functional capacity, which remains a cornerstone of diagnosis.

Interestingly, Ortega-Robles et al. [50] and Dal Pasquale et al. [51] reported that functional symptoms frequently coexist with mild physical abnormalities, leading to diagnostic confusion, particularly in older adults or those with a history of neurological injury.

#### 3.2.2. Diagnosis

Accurate diagnosis remains critical in FMD, and multiple articles evaluated the utility of positive diagnostic signs such as Hoover’s sign for functional weakness or entrainment in functional tremor [47,49,50,52]. Alluri et al. examined diagnostic accuracy in a large academic center, finding that expert clinical judgment, combined with positive signs, led to high interrater reliability in diagnosis [49].

Studies such as those by Garcin and LaFaver [53,54] emphasized the importance of standardized neurological examination and functional movement assessment protocols, encouraging clinicians to move away from diagnosis by exclusion. Many authors advocated for early diagnosis to prevent unnecessary investigations, prolonged disability, and healthcare overuse.

#### 3.2.3. Neurobiological Mechanisms

A growing body of work has investigated the neural correlations of FMD, particularly using functional MRI (fMRI), diffusion tensor imaging (DTI), and resting-state connectivity analyses [33,47,52,53,55].

For example, Huepe-Artigas et al. [48] identified altered connectivity in the limbic-motor interface, implicating regions like the insula, amygdala, and supplementary motor area (SMA) in the pathophysiology of FMD. Similarly, Edwards and colleagues reviewed evidence pointing to aberrant motor attention, reduced inhibition, and impaired agency as key neural signatures [5].

Multiple studies noted abnormal interactions between emotion regulation circuits and motor execution systems, reinforcing the biopsychosocial model of FMD [47,55,56].

#### 3.2.4. Psychiatric Comorbidities

Psychiatric symptoms—including anxiety, depression, PTSD, and dissociation—were consistently reported across FMD studies [49,53,54,57]. Several articles linked FMD onset to preceding trauma, acute stress, or interpersonal conflict, aligning with broader findings in functional neurological disorders [48,54].

Dal Pasquale et al. [51] categorized neuropsychiatric phenotypes in FMD, demonstrating that different psychiatric profiles correlate with symptom types and prognosis. Furthermore, Kola et al. [58] reported that untreated comorbid anxiety and depression significantly impair recovery in FMD, especially in chronic cases.

#### 3.2.5. Treatment

Treatment studies emphasized multidisciplinary approaches combining neurologic explanation, physiotherapy, cognitive behavioral therapy (CBT), and psychiatric care [47,53,59,60]. LaFaver et al. provided consensus-based treatment recommendations, advocating for early education, symptom validation, and functional restoration as primary therapeutic goals [54].

Innovative therapeutic models included retraining-based physiotherapy, as explored by Nielsen et al. and others [56,60], focusing on normalizing movement patterns rather than reinforcing disability-focused behaviors. Psychologically informed physical therapy was particularly effective for functional gait disorders and tremors.

In addition, new intervention strategies combining transdiagnostic CBT and trauma-focused therapy are being explored, with early pilot data suggesting benefits in symptom reduction and quality of life [53,54].

### 3.3. Functional Neurological Disorder (FND)

Functional Neurological Disorder (FND), historically referred to as conversion disorder, encompasses a broad spectrum of neurological symptoms that are inconsistent with known neurological disease. Forty-two articles in the dataset addressed general or mixed presentations of FND, spanning diverse topics such as epidemiology, clinical features, etiology, comorbidities, pathophysiology, and treatment.

#### 3.3.1. Clinical Profiles and Phenomenology

Several studies described the broad clinical heterogeneity of FND, including motor, sensory, cognitive, visual, and mixed presentations [61,62,63,64,65]. These diverse symptoms often overlap in the same patient, complicating diagnosis and care pathways. Miranda et al. highlighted the co-occurrence of motor and sensory symptoms in pediatric populations, emphasizing the challenge of distinguishing FND from rare inherited conditions [66].

Kaplan [67] and Domingos et al. [68] discussed the historical evolution of symptom classification in FND, noting that modern diagnostic frameworks now support a positive, rule-in approach rather than exclusion.

#### 3.3.2. Etiology and Risk Factors

Many articles explored predisposing and precipitating factors for FND. Adverse childhood experiences, psychological trauma, and chronic stress emerged as consistent contributors to vulnerability [69,70,71,72]. Zelinski et al. [73] identified altered brain–body integration in patients with FND, proposing that dysfunctional salience processing may underline the tendency to somatize stress.

In a psychodynamic review, Kaplan [67] framed FND as an expression of unconscious conflict and affect dysregulation, calling for better integration of historical models with current neuroscientific insights. Meanwhile, Alanazi et al. and others [61,74] found high rates of psychiatric comorbidities, especially anxiety and dissociation.

#### 3.3.3. Neurobiological Mechanisms

The neurobiology of FND remains a fast-evolving area, with multiple studies using fMRI and network connectivity models to propose mechanisms. Studies by Zelinski and colleagues [73,75,76] emphasized abnormalities in the default mode, salience, and sensorimotor networks—suggesting impaired self-agency, prediction errors, and motor inhibition.

Critically, these neural alterations are not static but appear to reflect state-dependent changes, further supporting the reversibility and functional nature of symptoms [76,77,78].

Others proposed altered interoceptive awareness and threat hypervigilance as underlying mechanisms, with activation of emotion-processing areas such as the amygdala and anterior cingulate cortex observed during motor or sensory symptom episodes [79,80,81].

#### 3.3.4. Diagnosis and Challenges

Articles by Stone, Bailey, and others addressed the challenge of diagnostic delay and mislabeling [33,77,79]. Patients with general FND often cycle through various specialties—neurology, psychiatry, gastroenterology—before receiving a correct diagnosis, contributing to frustration and mistrust.

Domingos et al. [68] and Miranda et al. [66] suggested that early, confident diagnosis by a neurologist using positive signs and patient-centered communication significantly improves acceptance and reduces unnecessary investigations.

#### 3.3.5. Treatment

Treatment was a major focus in over a third of the studies, most emphasizing multidisciplinary care models [65,70,72,82,83]. Cognitive behavioral therapy (CBT), trauma-focused therapy, functional physiotherapy, and education were central components.

LaFrance and colleagues [38,77] proposed a stepped-care model combining psychoeducation, psychological therapy, and coordinated neurological follow-up. Studies also supported symptom-focused physiotherapy as effective, particularly when guided by experienced therapists with an understanding of FND-specific movement patterns [27,84,85].

Psychiatric comorbidities were frequently reported as barriers to successful treatment, especially when unrecognized [74,81]. Innovative interventions included virtual reality therapy [86], group therapy formats [83], and family-based interventions in pediatric cases [87,88].

### 3.4. Functional Cognitive Disorder (FCD)

Two articles in the dataset addressed Functional Cognitive Disorder (FCD), a condition defined by cognitive complaints such as forgetfulness and attentional lapses that are not explained by neurodegeneration or structural brain disease.

Wackym et al. [89] emphasized the challenge of distinguishing FCD from early-stage dementia and mild cognitive impairment, especially in older adults. The study described common features of FCD, including excessive worry about cognition, subjective complaints not matching objective testing, and frequent co-occurrence with anxiety or depression. They recommended integrating neuropsychological testing with clinical interviews to improve diagnostic accuracy and avoid mislabeling.

Cabreira et al. [90] provided further insight by exploring precipitating factors in FCD. Their observational study identified psychosocial stressors, trauma exposure, and functional somatic syndromes as common antecedents. The authors proposed a stress-related model of FCD, where attentional bias, maladaptive beliefs about memory, and heightened self-monitoring contribute to persistent cognitive symptoms. Their work supports the classification of FCD within the broader spectrum of functional neurological disorders, with overlapping psychological and neurobiological mechanisms.

#### 3.4.1. Comparative Analysis of Comorbidity Profiles Across Functional Neurological Disorder Subtypes

Functional Neurological Disorders (FND) are characterized by a wide array of neurological symptoms that are incongruent with known structural pathology. However, comorbid conditions—particularly psychiatric, somatic, and cognitive—frequently accompany these presentations, influencing both symptomatology and prognosis. A comparative synthesis of comorbidity profiles across four major FND subtypes—Psychogenic Non-Epileptic Seizures (PNES), Motor FND (FMD), General or Mixed FND, and Functional Cognitive Disorder (FCD)—highlights important overlaps and distinctions that can guide clinical management [91].

#### 3.4.2. Psychiatric Comorbidities

Psychiatric conditions are prevalent across all FND subtypes, though they manifest with varying prominence and clinical implications. In PNES, psychiatric comorbidities are especially dominant, with frequent diagnoses of major depressive disorder, generalized anxiety disorder, post-traumatic stress disorder (PTSD), dissociative disorders, and personality disorders. These conditions often preceded seizure onset and significantly impact treatment outcomes [92].

In contrast, FMD studies emphasize anxiety, depression, PTSD, and history of trauma, but personality pathology is less frequently characterized. Psychiatric comorbidities in general FND exhibit a broader scope, often coexisting with somatization, dissociation, and stress-related disorders, reflecting the heterogeneity of symptom clusters [78]. Finally, FCD is frequently associated with anxiety and depressive disorders, but also displays a unique cognitive anxiety pattern, characterized by hypervigilance about mental performance, often without objective deficits [93].

#### 3.4.3. Somatic and Pain Comorbidities

Somatic symptoms and chronic pain are central to the PNES and FMD comorbidity spectrum. Chronic pain, fibromyalgia, fatigue syndromes, and gastrointestinal disturbances are commonly reported in PNES, sometimes overshadowing seizure-like episodes and leading to diagnostic confusion [94]. FMD also demonstrates significant overlap with pain syndromes, including functional pain disorders and fatigue, particularly in patients with coexisting gait disturbances and dystonia.

In general, FND, functional somatic syndromes (e.g., non-cardiac chest pain, irritable bowel syndrome) frequently co-occur, reinforcing the concept of shared pathophysiological mechanisms such as central sensitization and altered interoception [95]. While FCD is primarily defined by cognitive complaints, the literature suggests occasional overlap with functional somatic symptoms, although this remains underexplored [96].

#### 3.4.4. Cognitive Comorbidities

Cognitive complaints vary in nature and salience across subtypes. In PNES, patients frequently report memory lapses, concentration difficulties, and executive dysfunction, often linked to underlying emotional distress [97]. FMD patients may describe brain fog, attentional bias, and reduced concentration, often correlated with the emotional burden of their motor symptoms [49].

General FND literature frequently references subjective cognitive dysfunction, though symptoms may fluctuate or appear context-dependent. Dissociation and attentional shifts often mediate these complaints, rather than clear structural or neurodegenerative changes [63].

FCD, by definition, centers on subjective cognitive deficits in memory and attention, despite intact objective testing. It is distinct in its metacognitive disruption, where patients display reduced confidence in memory, excessive self-monitoring, and catastrophic interpretations of normal cognitive lapses. Comorbid anxiety and trauma are known to exacerbate this profile [72].

To contextualize the findings and facilitate clinical translation, Table 2 outlines key implications and therapeutic recommendations specific to each FND subtype.

## 4. Discussion

The results of this comprehensive analysis highlight the pivotal role of comorbidities in shaping the clinical presentation, diagnostic complexity, and therapeutic responsiveness across subtypes of Functional Neurological Disorder (FND). Rather than being incidental findings, psychiatric and somatic comorbidities emerge as core elements of the disorder’s expression. Their prevalence varies by subtype, yet they consistently influence symptom formation, illness beliefs, and the overall patient journey [98].

Functional neuroimaging has played a key role in redefining FND as a disorder of brain network dysfunction rather than structural damage. Studies using functional MRI (fMRI), PET, and SPECT have demonstrated altered connectivity in several brain networks, notably the salience network, default mode network, and limbic-motor circuits. Regions such as the anterior cingulate cortex, insula, amygdala, and supplementary motor area (SMA) are consistently implicated. In PNES, functional imaging often reveals increased activation in emotion-processing and dissociative circuits, particularly during seizure-like events. By contrast, motor FND is associated with impaired connectivity between motor intention areas and execution regions, reflecting disturbances in agency. These distinct patterns support the view that FND subtypes, while sharing core mechanisms, exhibit unique neurofunctional profiles, knowledge that may aid differential diagnosis and therapeutic targeting [4,13,32,47].

Symptom-based categorization of FND subtypes offers important insight into pathophysiology and therapeutic needs. Functional motor symptoms, such as limb weakness or tremors, often reflect disrupted motor planning and agency, with strong involvement of sensorimotor and frontal-parietal circuits. Disorders involving speech, such as functional aphonia, dysarthria, or stuttering, frequently co-occur with anxiety and may engage limbic-laryngeal pathways and altered connectivity in language-dominant cortices. In contrast, functional pain syndromes and chronic headaches are frequently linked to heightened interoceptive awareness and central sensitization mechanisms, overlapping with somatic symptom disorders. Management strategies vary motor symptoms that benefit from physiotherapy and motor retraining, speech impairments may respond to speech therapy combined with CBT, and chronic pain or headache requires integrated pain management, including behavioral approaches, relaxation training, and in some cases, pharmacological support. This differentiation underscores the need for individualized, symptom-specific treatment planning [14,28,41,48].

Given the complex interplay of neurological, psychological, and somatic factors in FND, a multidisciplinary approach is essential. Optimal management should involve close collaboration between neurologists, psychiatrists, psychologists, physiotherapists, occupational therapists, and, where indicated, speech and language therapists. Early, coordinated care facilitates accurate diagnosis, addresses comorbidities, and supports functional restoration. Evidence suggests that multidisciplinary rehabilitation programs tailored to the dominant symptom profile, motor retraining for movement disorders, trauma-focused therapy for PNES, and metacognitive training for FCD, are associated with better functional outcomes and lower relapse rates.

In the case of Functional Cognitive Disorder (FCD), psychiatric conditions, particularly anxiety disorders, depression, and health-related anxiety, frequently dominate the clinical picture. Our findings reveal that anxiety was present in over 48% of FCD cases, with depression and fatigue following closely. These comorbidities do not merely coexist with cognitive complaints but are deeply intertwined with the phenomenology of the disorder. Patients with FCD often report debilitating forgetfulness, poor concentration, or difficulties with word retrieval. When these cognitive symptoms arise amidst psychological distress, attentional bias toward perceived cognitive deficits and diminished cognitive confidence can create a reinforcing loop. Moreover, comorbid somatic conditions such as sleep disturbance, fibromyalgia, and migraine are common and often reinforce illness convictions, complicating the diagnostic distinction between functional cognitive changes and early neurodegenerative processes. This diagnostic ambiguity is particularly concerning in older patients, where the fear of dementia may color both patient narratives and clinician suspicion, often resulting in extended investigations or delayed functional diagnosis [27,72]. From a treatment perspective, these findings point to the necessity of dual-pronged approaches that address both the cognitive and emotional dimensions of the disorder.

Functional Movement Disorder (FMD) emerged in this study as the subtype most consistently associated with a high overall burden of comorbidities. Psychiatric comorbidities were strikingly prevalent, with depression present in over 70% of cases and anxiety exceeding 80% in several datasets. Early life trauma and adverse childhood experiences were also frequently reported, suggesting a significant role for stress and psychological vulnerability in shaping motor symptomatology [8]. Fatigue, chronic pain, and migraine were among the most common physical comorbidities. These symptoms frequently co-occurred in complex constellations, contributing to a high comorbidity load that both mirrors and exacerbates the motor manifestations of FMD. Our comparative analysis shows that patients with FMD not only report multiple comorbid symptoms but often present with a functional overlay that can obscure primary motor diagnoses such as Parkinsonism or dystonia. The presence of these symptoms may also impair engagement with rehabilitation, particularly when pain and fatigue reduce physical stamina or increase activity avoidance. Therefore, these findings underscore the importance of a personalized, interdisciplinary model of care that incorporates pain management, physiotherapy, psychological support, and trauma-informed intervention [99].

For Psychogenic Non-Epileptic Seizures (PNES), psychiatric comorbidities are almost universally present and are critical to both diagnosis and management. Our data show that PTSD and dissociative disorders, including borderline personality features, were reported in 55–70% of patients across studies, while depression and anxiety were consistently observed in over half of all cases. In many instances, these psychiatric comorbidities predate the onset of PNES and contribute directly to dissociative seizure mechanisms [100,101,102]. Trauma history, including childhood abuse and neglect, was frequently cited as a significant etiological factor. Somatic conditions were also well-represented, with fibromyalgia, irritable bowel syndrome, and chronic pelvic pain noted in multiple studies [103,104]. These overlapping complaints contribute to central sensitization, reinforcing the functional symptom pattern [105]. Diagnostic clarity in PNES is often complicated by the presence of these comorbidities, which may mask the functional nature of episodes or confuse them with epileptic seizures. However, recognizing the prominence of psychiatric pathology in PNES can also support diagnostic certainty, especially when seizure semiology is ambiguous. Nonetheless, stigma and misunderstanding remain significant barriers [106,107]. Many patients face skepticism from clinicians, particularly when psychiatric comorbidities dominate the clinical picture, underlining the importance of empathetic communication and transparent diagnostic framing [108,109].

Persistent Postural-Perceptual Dizziness (PPPD) represents another clinically significant subtype, characterized by a unique interaction between vestibular, psychological, and autonomic factors. Although fewer studies addressed PPPD directly, those that did consistently reported high rates of anxiety spectrum disorders, especially panic disorder and generalized anxiety. Chronic dizziness in PPPD is often perpetuated not only by vestibular instability but also by hypervigilance to bodily sensations, increased postural focus, and maladaptive cognitive schemas about balance and movement [110]. Our findings emphasize that PPPD cannot be managed effectively through vestibular rehabilitation alone. Comorbid anxiety must be addressed concurrently, ideally through integrated cognitive-behavioral therapy that recalibrates attentional mechanisms and reduces safety-seeking behaviors. Moreover, comorbid conditions such as migraine and orthostatic intolerance were also observed, further complicating the clinical picture and highlighting the importance of a transdiagnostic approach [111,112].

Across all FND subtypes, our cluster and correlation analyses revealed frequent co-occurrence of specific comorbidities, such as the pairing of anxiety and depression, or fatigue and chronic pain [113,114]. These combinations often form distinct clinical phenotypes that transcend traditional subtype boundaries. Such comorbidity patterns have substantial implications for both diagnosis and treatment. Patients with high comorbidity burdens tend to experience longer symptom duration, increased functional impairment, and reduced responsiveness to single-modality interventions. Furthermore, comorbid conditions often shape how patients interpret their symptoms. For instance, individuals with fibromyalgia or chronic fatigue may interpret new neurological symptoms as further evidence of an organic or degenerative illness, thereby reinforce maladaptive illness beliefs and increasing resistance to the functional formulation.

Our comparative statistical analysis supports these findings, revealing significant differences in the distribution and clustering of comorbidities across FND subtypes. FMD was associated with the highest number of coexisting conditions, while FCD displayed a more focused psychiatric profile with fewer physical complaints [115,116]. Chi-square testing demonstrated that PTSD and dissociation were significantly more prevalent in PNES than in other subtypes, while fatigue and chronic pain were especially elevated in FMD. These patterns suggest that diagnostic and treatment strategies must be tailored to subtype-specific comorbidity profiles. Clinicians must also remain vigilant for the risk of diagnostic overshadowing, where attention to comorbid psychiatric symptoms may lead to under-recognition of the functional neurological presentation, or conversely, where functional symptoms obscure serious coexisting medical conditions [117,118,119].

Despite the comprehensive nature of this review, certain limitations must be acknowledged. The heterogeneity in study designs, diagnostic criteria, and assessment methods across included studies limits direct comparison and synthesis. Some FND subtypes, particularly PPPD and FCD, remain underrepresented, potentially biasing the overall conclusions. Furthermore, cultural and ethnic diversity is insufficiently addressed in the literature, restricting the generalizability of findings. Lastly, as a narrative rather than systematic review, this synthesis may be subject to selection and publication biases.

Future research directions should aim to address these gaps by developing standardized diagnostic and comorbidity assessment tools tailored to diverse populations and FND subtypes. Longitudinal studies are needed to elucidate the causal relationship between psychiatric and somatic comorbidities and FND symptom trajectories, as well as to identify biomarkers that may aid in differential diagnosis. Moreover, neuroimaging research should continue to refine subtype-specific brain networks models, incorporating multimodal imaging and functional connectivity analyses to better understand underlying pathophysiology. Interventional studies are crucial to evaluate the efficacy of personalized, multidisciplinary treatment approaches, including novel behavioral therapies and pharmacological agents targeting comorbidities. Special attention should be given to understudied subtypes such as PPPD and FCD, as well as culturally diverse cohorts, to improve generalizability and equity in care. Finally, integrating patient perspectives and illness beliefs into research design could enhance therapeutic engagement and outcome measures.

## 5. Conclusions

In conclusion, functional neurological disorder (FND) reveals its complexity only when viewed through a biopsychosocial prism: biological predisposition, traumatic events and social context intertwine to generate a diverse range of clinical manifestations. Psychiatric comorbidities such as anxiety, depression and PTSD, as well as somatic comorbidities, chronic pain, fatigue or sleep disorders, are not simple coincidences, but active partners in the maintenance of functional symptoms. This subtle interaction explains why in psychogenic non-epileptic epilepsy, the burden of trauma often precedes seizures, why in functional motor disorders feelings of pain and exhaustion make the rehabilitation process difficult and why in functional cognitive disorders, cognitive anxiety amplifies the feeling of “mental fog” and the avoidance of challenging situations.

Across all subtypes, the high prevalence of psychiatric and somatic comorbidities is a defining feature of FND, with direct implications for prognosis and therapeutic planning. Their systematic identification should be considered a priority in both research and routine clinical practice, as failure to address them adequately can limit treatment response and increase the risk of relapse.

Early, comprehensive, and multidimensional assessment thus becomes the cornerstone of any therapeutic plan: a careful trauma history, specific dissociation and quantification scales for pain, fatigue, and sleep disorders, as well as detailed neuropsychological assessment can quickly identify vulnerable areas and direct the patient to the most appropriate services. In the absence of this early triage, the diagnosis risks remaining fragmented, and the patient caught between specialties, may be left without a clear path to recovery.

In practice, treatment must adapt to each comorbidity profile: from trauma-focused therapies, such as trauma-focused CBT or EMDR, for people with histories marked by adverse events, to multidisciplinary rehabilitation programs that combine graded exercises and cognitive strategies for pain and fatigue management, and to metacognitive interventions and attention and memory training exercises for those affected by functional cognitive disorders. In all these situations, validating symptoms and establishing common goals strengthen the therapeutic alliance and increase adherence to treatment.

In the long term, advancing knowledge and the effectiveness of interventions will depend on research efforts to longitudinally map the neural circuits involved in NFD and how they respond to therapies, identify peripheral and neurophysiological biomarkers capable of anticipating relapse risk, and test therapeutic protocols stratified by comorbidity profile.

An additional critical aspect requiring increased attention is the insufficient representation of cultural and ethnic diversity in current FND research. Tailoring diagnostic and treatment strategies to the specific needs of diverse populations, including African Americans, Pacific Islanders, Asians, Caucasians, and Europeans, is essential to ensure equity and clinical effectiveness. Future studies should incorporate diverse cohorts and investigate potential biological, psychosocial. and cultural differences to personalize interventions and reduce disparities in access and therapeutic outcomes.

In parallel, integrating healthcare services through clear triage pathways and outcome monitoring will reduce fragmentation of care and provide a sustainable perspective on patient prognosis. Only by responsibly embracing the biopsychosocial model, by valuing comorbidities as pieces of the puzzle, and by building a collaborative and person-centered framework for research and clinical practice will we be able to transform the historical challenges of NFD into stories of recovery and lasting resilience.

Future research should focus on standardizing diagnostic criteria and comorbidity assessments to enhance comparability across studies and clinical settings. Longitudinal and interventional studies are needed to clarify the causal relationship and test targeted, multidisciplinary treatments tailored to subtype-specific profiles. Additionally, expanding research into underexplored subtypes and diverse populations will improve understanding and equity in clinical care for FND.

## Figures and Tables

**Figure 1 life-15-01322-f001:**
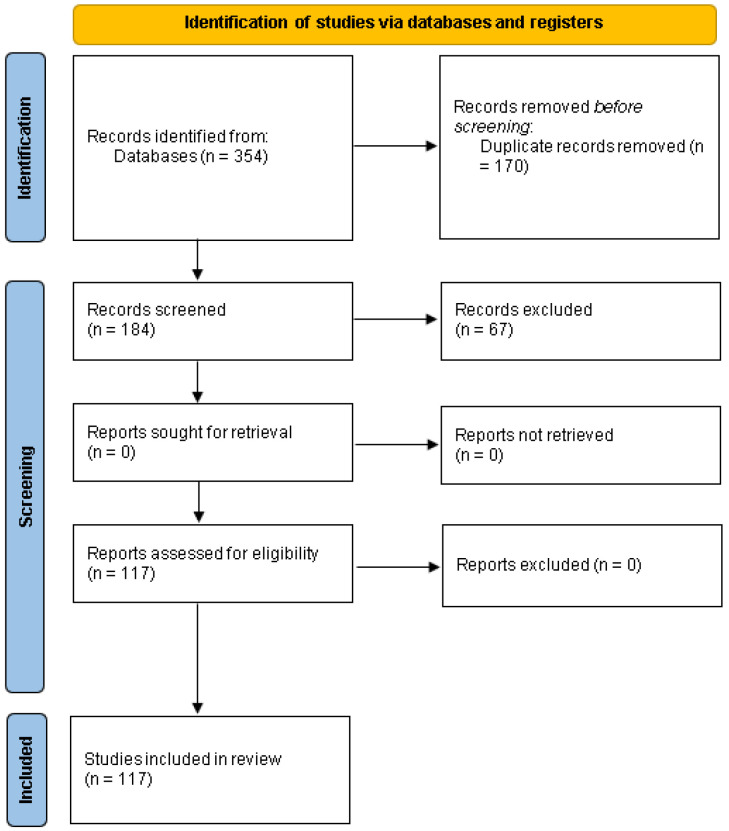
PRISMA 2020 flow diagram, which included searches of databases and registers only, according to the guidelines provided by PRISMA [12].

**Table 1 life-15-01322-t001:** Comorbid Psychiatric and Somatic Conditions Across FND Subtypes.

FND Subtype	Predominant Psychiatric Comorbidities	Frequent Somatic Comorbidities	Key Clinical Characteristics
PNES (Psychogenic Non-Epileptic Seizures)	PTSD, major depressive disorder, personality disorder	Migraine, chronic pain, fibromyalgia	Often linked with trauma history and misdiagnosis
FMD (Functional Motor Disorders)	Anxiety disorders, depression	Chronic musculoskeletal pain	Gait disturbances, weakness, tremor
FCD (Functional Cognitive Disorder)	Panic disorder, health anxiety, depression	Headache, fatigue	Memory complaints, attention deficits
PPPD (Persistent Postural-perceptual Dizziness)	Phobic disorders, generalized anxiety	Vestibular dysfunctions	Chronic dizziness, postural instability

**Table 2 life-15-01322-t002:** Clinical Implications and Therapeutic Suggestions by FND Subtype.

FND Subtypes	Clinical Implications	Recommended Therapeutic Approaches
PNES	Psychiatric comorbidity burden, delayed diagnosis	Trauma-focused CBT, psychoeducation, psychiatric support
FMD	Functional disability, persistent symptoms	Multidisciplinary rehabilitation, physiotherapy
FCD	Cognitive misattribution, excessive health worry	Cognitive restructuring, patient education
PPPD	Anxiety-vestibular interplay, chronic dizziness	Vestibular rehabilitation, SSRIs, CBT

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
