# Peer review of "Comorbidities Across Functional Neurological Disorder Subtypes: A Comprehensive Narrative Synthesis"

_life, 2025, doi:10.3390/life15081322_

Round 1

Reviewer 1 Report

Comments and Suggestions for Authors

The study has value and that it is focusing in an area of functional neurologic disorder that does require some review so it is scientifically relevant.

The literature review need some work in terms of reviewing functional image findings for functional neurologic disorder and delineating it from other types of neurological disorders that can be functional such as psychogenic seizures, or non-epileptic psychogenic seizures.

would want a discussion with findings on functional imaging and the differences in these two disorders also the difference in neurological disorders involving speech versus weakness of a body part versus chronic headaches or chronic pain and what are the differences among these types of functional disorders and how do you treat these functional disorders.

Terms of results and I would recommend a focus also on the multidisciplinary approach to functional neurological disorder and how you would recommend approaching caring for these patients.

In terms of relevance to clinicians, we would recommend focusing on how to study is relevant to treating different populations with functional disorders, such as African-Americans , Pacific Islanders and those of Asian heritage and those Caucasian heritage and those of European heritage.

If the literature is lacking in these areas which we suspect it is lacking, then a brief discussion on the limitations of current therapy should be included.

Generally, a good review and with minor changes would recommend publication.

Reviewer 2 Report

Comments and Suggestions for Authors

Abstract: Appropriate, aims are clearly descrived, results are well structured, conclusion is useful.

p. 26. Please provide a time period for the included papers.

Introduction: Instead of the first sentence, pleae provide a puncutal definition of FND, e.g., https://www.nhsinform.scot/illnesses-and-conditions/brain-nerves-and-spinal-cord/functional-neurological-disorder/ or https://www.ninds.nih.gov/health-information/disorders/functional-neurologic-disorder

Also, please describe the subtypes of FND and some in sight into etiology (e.g., genetic, viral origin etc. )

Method: Appropritate, but:

pp. 92-104, the research gap and objectives are clearly described.

p. 120: please also provide the publication period for the inluded resources. 

Results: Clealy desribed and well structured, easy to follow.

Discussion

Comorbid states are really common. I recommend that the authors should better emphasize this evidence as well as the possible causes and clinical consequences of this.

Also, please describe the strengths of and limitations to the present study, and give some directions for future research.

Conclusion

Again the high prevalence of comorbid states should be mentioned.

Overall, this is a well written and useful paper of high quality.

Reviewer 3 Report

Comments and Suggestions for Authors

Please address the following aspects before resubmitting a revised version of the manuscript:

  1. While you state your work is a comprehensive narrative synthesis, you include in the materials and methods section a PRISMA diagram from a systematic review. You should be very clear about what type of paper you want to prepare.
  2. The introduction section should be improved. As this is a general purpose journal for a large audience, I suggest you give some definitions (and examples) for the terms you use: Functional Cognitive Disorder (FCD), Functional Movement Disorder (FMD), and Psychogenic Non-Epileptic Seizures (PNES).
  3. The results section could be improved and made more reader-friendly by adding 2-3 summarizing tables.
  4. In the final part of the discussion section, I suggest you add a short paragraph on the limitations of your review and the currently existing literature. 
  5. Please suggest 2-3 future research directions in the final part of the conclusion section. 
